# MeJA Elicitation on Flavonoid Biosynthesis and Gene Expression in the Hairy Roots of *Glycyrrhiza glabra* L.

**DOI:** 10.3390/genes16111387

**Published:** 2025-11-18

**Authors:** Yutao Zhu, Bohan Wang, Bingyi Xue, Runqian Wang, Ganlin Tang, Tao Zhu, Mei Zhao, Taotao Li, Chunli Liao, Huamin Zhang, Dongxiao Liu, Jianhua Chen, Lianzhe Wang

**Affiliations:** 1College of Life Science and Engineering, Henan University of Urban Construction, Pingdingshan 467036, China; wbh20050323@163.com (B.W.); xuebingyi2024@163.com (B.X.); 18339532937@163.com (R.W.); 15753972901@163.com (G.T.); 30110201@hncj.edu.cn (T.Z.); meizhao2021@163.com (M.Z.); ltt198906@163.com (T.L.); liao20130427@163.com (C.L.); hmzhang111@126.com (H.Z.); 20221095@hncj.edu.cn (D.L.); jijy99@126.com (L.W.); 2Pingdingshan Academy of Agricultural Sciences, Pingdingshan 467003, China; cjh8512@sina.com

**Keywords:** *Agrobacterium rhizogenes*, *Glycyrrhiza glabra*, hairy roots, flavonoid biosynthesis, transcriptome analysis

## Abstract

**Background/Objectives:** Licorice (*Glycyrrhiza glabra* L.) is a highly important medicinal plant that is widely used in China owing to its active ingredients. Its main active components are flavonoids, including liquiritigenin, liquiritin and licochalcone A. The hairy roots (HRs) induced by *Agrobacterium rhizogenes* are a commonly used chassis in synthetic biology to enhance the production of active compounds in medicinal plants. **Methods:** A biosynthesis system to acquire the active ingredients of *G. glabra* was established using an HR culture system. It employed a transcriptome analysis to identify the change in gene expression following treatment with methyl jasmonate (MeJA). **Results:** After 28 days of suspension culture, the biomass of HRs increased by approximately 34.5-fold and reached 1.83 g/100 mL flask. Treatment with MeJA significantly increased the contents of liquiritigenin, liquiritin, and glabridin in the HRs. The transcriptome data indicated that MeJA activated the flavonoid biosynthetic pathway genes in the HRs, which was largely consistent with the qRT-PCR results. Furthermore, the overexpression of the *GgCHS6* gene substantially increased the content of flavonoids in HRs. **Conclusions:** Collectively, this study established an HR system to biosynthesize the active ingredients of *G. glabra* using metabolic engineering and genetic engineering techniques and provides several valuable candidate genes for further functional study.

## 1. Introduction

Licorice is the dried root and rhizome of *Glycyrrhiza uralensis* Fisch., *Glycyrrhiza inflata* Bat., or *Glycyrrhiza glabra* L., which are plants in the legume family; it has been used medicinally for thousands of years in China [1]. The flavonoids in licorice are more diverse, and the primary pharmacologically active compounds include liquiritigenin, isoliquiritigenin, liquiritin, licochalcone A, and glabridin [2]. However, the reduction in wild licorice resources and the decline in the quality of cultivated varieties have severely constrained the acquisition of medicinal components derived from this plant. Given the critical importance of the active ingredients of licorice in medical and food applications, their rapid and efficient biosynthesis through synthetic biology methods is highly significant and of practical value.

HRs are types of root-like tissues that are produced at a wound site after plants have been infected by *Agrobacterium rhizogenes* [3]. Owing to their advantages, such as a short growth cycle, high degree of differentiation, hormone-free culture, genetic stability, and ease of manipulation, HRs have become a commonly used plant chassis for the biosynthesis of plant secondary metabolites [4,5,6]. There are more than 200 medicinal plants that have established HR culture systems for the biosynthesis of saponins, terpenoids, phenylpropanoids, alkaloids, flavonoids, anthraquinones, and polysaccharides [7]. There are typically economically valuable plant secondary metabolites at low concentrations in both plant tissues and HRs under natural growth conditions. Altering the culture environment, such as the type of light and its intensity, temperature, and rotation speed, can enhance the active components in HRs. For example, Afsharzadeh et al. demonstrated that the use of red-blue composite light significantly increased the content of total phenolic in HRs of *G. glabra* [8]. In addition, the incorporation of exogenous plant hormones, chitin, chitosan, and metal particles as elicitors in the HR cultivation system can also significantly enhance the content of plant secondary metabolites in HRs [7]. Among the commonly used in vitro elicitors, the plant hormone methyl jasmonate (MeJA) has been widely employed to dramatically increase the production of plant secondary metabolites [9,10].

Numerous studies have shown that the exogenous application of MeJA can enhance the content of secondary metabolites in HRs. Wongwicha et al. (2011) demonstrated that the addition of MeJA to the HR culture system of *G. inflata* significantly increased the content of glycyrrhizin [11]. MeJA can significantly promote the gene expression of genes for the tanshinone and phenolic acid biosynthetic pathways in the HRs of *Salvia przewalskii* Maxim., thus leading to a substantial increase in the contents of caffeic acid, salvianolic acid B, and tanshinone IIA [12]. Similarly, the content of total phenolics and flavonoids in *Ficus carica*. Cv. Siah HRs were increased by MeJA elicitation, which may be the result of a significant enhancement in the expression of chalcone synthase (CHS) and flavonoid 3-hydroxylase (F3H) genes [13]. In addition, Bao et al. (2022) found that MeJA treatment of broccoli (*Brassica oleracea* var. *italica*) HRs activated the expression of the key genes that regulate the biosynthesis of sulforaphane, which significantly enhances its production [14].

Flavonoids are the primary active components in licorice, and they possess a variety of significant pharmacological activities. In this study, the addition of MeJA can promote an increase in the contents of isoliquiritigenin, liquiritigenin, liquiritin, licochalcone A, and glabridin in the HRs, while salicylic acid (SA) was less effective at promoting such effects. A transcriptome analysis revealed that treatment with MeJA activated the flavonoid biosynthetic pathway in the HRs, and quantitative reverse-transcription PCR (qRT-PCR) data confirmed the significant upregulation of key flavonoid biosynthetic genes. Furthermore, the overexpression of *GgCHS6* substantially enhanced the levels of flavonoids in the HRs. Collectively, this study provides a new perspective to explore the complex molecular mechanism of the accumulation of flavonoids mediated by MeJA and provides several valuable candidate genes for further functional study.

## 2. Materials and Methods

### 2.1. Plant Materials

*G. glabra* seeds were collected from Altay, Xinjiang, China. Plump seeds were soaked in concentrated sulfuric acid for 60 min, thoroughly rinsed with tap water, and then transferred to sterile centrifuge tubes. Subsequent sterile washes included the following: (1) three rinses with sterile water, (2) 10 min of disinfection in 75% ethanol, (3) 10 min treatment with 10% sodium hypochlorite, and (4) three final sterile water rinses. Finally, the sterilized seeds were then cultured on 1/2 MS solid medium in a growth chamber maintained at 25 °C with a 16/8 h (light/dark) photoperiod.

### 2.2. Inoculation of A. rhizogenes K599

This study used the *A. rhizogenes* K599 strain, which is resistant to streptomycin and harbors the Ri plasmid pRi2659, to induce the HRs [15]. A volume of 1 mL of activated K599 bacteria was added to 100 mL of LB liquid medium that contained 50 μg/mL streptomycin. The culture was then incubated at 28 °C and 200 rpm in the dark in a shaker for 24–48 h. The activated culture was then centrifuged and resuspended in a 1/2 MS liquid medium that contained 100 μmol/L acetosyringone (AS), with an OD_600_ that ranged from 0.4 to 1.0.

### 2.3. Induction and Parameter Optimization of the HRs

The roots of 5-day-old *G. glabra* seedlings were excised to leave the hypocotyls with cotyledons as explants. The explants were immersed in a suspension of K599 for 10 min and then placed on 1/2 MS solid medium that contained 100 μM AS for culture in the dark for 2 d. After 2 d, the explants were rinsed with 1/2 MS liquid medium that contained 500 mg/L Timentin and transferred to 1/2 MS solid medium that contained 500 mg/L Timentin for induction culture. The induction rate of HRs was counted after 7 d [16]. Various parameters, such as the density of bacterial suspension (OD_600_ of 0.4, 0.6, 0.8, and 1.0), infection time (5, 10, 15, and 20 min), types of culture media (MS, 1/2 MS, 6,7-V, and B5), and concentration of AS (50, 100, 150, and 200 μmol/L), were studied to determine the optimal conditions to induce the HRs in *G. glabra*.

### 2.4. PCR Analysis to Screen the Transgenic HRs

The T-DNA that contained the *rolB* and *rolC* was responsible for the induction of HRs in plants. The successfully transformed HRs were screened by extracting the genomic DNA of the HRs using the CTAB method, followed by PCR analysis to detect the presence of *rolB* and *rolC* genes in the T-DNA (GenBank: Z29365.1) [17,18,19]. Additionally, the *VirG* gene was also detected to confirm the absence of *A. rhizogenes* in the HRs [20]. The primers are listed in Appendix A. The PCR was conducted by amplification with an initial denaturation at 95 °C for 3 min, followed by 32 cycles of denaturation at 94 °C for 25 s, annealing at 57 °C for 25 s (*rolB* at 57 °C, *rolC* at 55 °C, and *VirG* at 56 °C), and extension at 72 °C for 15 s with a final extension of 72 °C for 5 min using a PCR instrument (Bio-Rad, Hercules, CA, USA).

### 2.5. Analysis of the Growth Kinetics of HRs

A total of 2–3 cm (approximately 50 mg fresh weight) of successfully transformed and young HR shoots were used to inoculate them into 100 mL of 1/2 MS liquid medium that contained 3% sucrose and 200 mg/L Timentin (using a 250 mL conical flask). All the cultures were incubated at 150 rpm on a rotary shaker at 28 ± 1 °C. The growth curve was analyzed by harvesting three independent fresh HRs at 7, 14, 21, 28, 35, and 42 days to determine their fresh weight (FW).

### 2.6. HRs Treatments by Elicitors

For the induction experiment, the HRs were inoculated into 100 mL of 1/2 MS liquid medium that contained 200 mg/L Timentin in 250 mL conical flasks for subculture to facilitate the subsequent induction treatments. The flasks were shaken continuously at 150 rpm on a rotary shaker and cultured in the dark at 28 ± 1 °C for 28 days. MeJA (Sigma-Aldrich, St. Louis, MO, USA) and salicylic acid (SA) (Sigma-Aldrich) were dissolved separately in absolute ethanol and 20% ethanol, respectively, to prepare 100 mM stock solutions, which were filter-sterilized and added to the medium to achieve a final concentration of 100 μM [21]. Samples were collected at 0, 3, and 5 days after the elicitor treatment, and there were three replicates for each treatment group. The harvested HRs were weighed to determine their FW, rapidly frozen in liquid nitrogen, and stored at −80 °C until further analysis.

### 2.7. Use of HPLC-MS to Quantify the Contents of Flavonoids in the HRs

The active compounds in HRs were detected as previously described [22,23] with modifications. In brief, 100 mg of freeze-dried HRs were vortexed to mix with 1 mL of extraction solvent (methanol/water = 4:1 [*v*/*v*]) for 5 min for extraction. Following 10 min of ultrasonication and centrifugation at 1000 rpm, the supernatant (1 mL) was purified using 100 mg of PSA and filtered through a 0.22 μm membrane into an autosampler vial. The contents of isoliquiritigenin, liquiritigenin, liquiritin, licochalcone A, and glabridin in the HRs were determined using an Agilent 6495D triple quadrupole liquid chromatography-mass spectrometry (LC-MS) system (Agilent 1290 Infinity II, Agilent Technologies, Santa Clara, CA, USA), which was equipped with a C18 column (2.1 × 50 mm, 1.8 µm). Mobile phases A and B were acetonitrile and 0.1% formic acid in water, respectively. A linear gradient elution program was employed as follows: 0 min: 5% A; 4 min: 95% A; 5 min: 95% A; 5.1 min: 5% A; and 6.1 min: 95% A with a flow rate of 0.2 mL/min. The injection volume was 2 μL. The mass spectrometry parameters of the flavonoid standards are shown in Appendix A.

### 2.8. Transcriptome Analysis and Gene Annotation

HRs treated with 100 μM MeJA were chosen for further transcriptome analysis. The HRs that had grown to day 28 were treated with 100 μM MeJA and harvested at 0, 3, and 5 d. There were three biological replicates for each treatment. The HRs were quickly frozen in liquid nitrogen and then stored at −80 °C until the transcriptome analysis. A total of 1 μg total RNA was obtained from the HRs and used to construct RNA-seq libraries. An Illumina HiSeq/Illumina Novaseq/MGI2000 instrument (Illumina, San Diego, CA, USA) with a 2 × 150 paired-end (PE) configuration was used for the high-throughput sequencing. The raw sequencing reads in the FASTQ format were processed using Cutadapt (v. 1.9.1) to remove the technical sequences and low-quality bases, which can introduce bias and reduce the reliability of analysis [24]. Clean reads were aligned to the *G. uralensis* Fisch. Reference genome [25] using HISAT2 (v. 2.2.1) [26]. A transcriptome assembly was performed using StringTie (v. 1.3.3b) [27], and the novel transcripts were identified by a comparison of the assembled transcripts to the reference annotation using cuffcompare (v. 2.2.1) [28]. Differentially expressed genes (DEGs) were identified using DESeq2 (v. 1.34.0) (FDR < 0.05 and |log_2_FC| ≥ 1) [29]. A functional enrichment analysis interprets the biological significance of DEGs by mapping them to the Gene Ontology (GO) terms [30] and Kyoto Encyclopedia of Genes and Genomes (KEGG) pathways [31]. The DEGs were clustered using K-means clustering of the DEGs based on their Fragments Per Kilobase of transcript per Million mapped reads (FPKM) values using the Mfuzz package in R [32].

### 2.9. Validation of Gene Expression by Quantitative Real-Time PCR

To verify the reliability of the RNA-seq data, an RT-qPCR expression analysis was used to identify candidate genes on the KEGG synthesis pathway, including *GgPAL3*, *GgCHS6*, *GgCHS7*, *GgCHI1*, *GgUGT*, *GgHI4OMT*, *GgCYP81E1*, *GgCYP81E*4, and *GgVR1*. An RNA extraction kit (Biomed Gene Technology, Co., Ltd., Beijing, China) was used to extract the total RNA from HRs. The cDNA was synthesized from 1000 ng total RNA using an M5 First Strand cDNA Synthesis Kit (Mei5 Biotechnology, Co., Ltd., Beijing, China). A qRT-PCR was conducted using 2 × M5 HiPer Realtime PCR Super mix with Low (SYBRgreen, with anti-Taq) (Mei5 Biotechnology, Co., Ltd.) on a QuantStudio 6 Pro system (Applied Biosystems, Waltham, MA, USA). Gene expression was normalized using actin (EU190972.1) as a reference gene to normalize the cDNA [33]. The relative gene expression was calculated using the 2^−ΔΔCt^ method [34]. All the qRT-PCR experiments were performed in three biological replicates. All the primers are listed in Appendix A.

### 2.10. Construction of GgCHS6 That Overexpresses the HRs

The coding sequence of the chalcone synthase gene *GgCHS6* with Nco *I* and Spe *I* was inserted into the pCAMBIA 1302 vector that harbored hygromycin phosphotransferase (hptII) and green fluorescent protein (GFP) under the control of the CaMV 35S promoter (Appendix A). The construct was then introduced into *A. rhizogenes* K599 by the freeze–thaw method [35], and the transgenic HRs were induced as described in Section 2.3. The HRs transformed with the 1302 empty vector were used as the control. A total of 2–3 cm HRs were inoculated into 100 mL of 1/2 MS liquid medium that contained 3% sucrose and 200 mg/L Timentin (using a 250 mL conical flask). After 28 d of cultivation, the HRs were harvested and analyzed by high-pressure liquid chromatography-mass spectrometry (HPLC-MS) to determine the contents of isoliquiritigenin, liquiritigenin, liquiritin, licochalcone A and glabridin.

### 2.11. Data Analysis

Data are obtained from three independent replicates per treatment and presented as mean ± standard deviation. Significant differences between groups were compared with a confidence level of *p* < 0.05 by Duncan’s multiple range test using statistical software IBM SPSS Statistics 20.

## 3. Results

### 3.1. Optimization of the Induction and Condition of the HRs

The hypocotyls (with cotyledons) of 5-day-old *G. glabra* were soaked in *A. rhizogenes* strain K599. The HRs could be observed at the wounded site as early as 2 d after the end of co-cultivation and elongated further at 5 d (Figure 1A,B). The *rol* genes, such as *rolB* and *rolC,* in the T-DNA region of the Ri plasmid, play a significant role in inducing the formation of HRs and are ultimately integrated into the plant cell genome [7]. Therefore, they are commonly used as reference genes to confirm the formation of HR using PCR. The presence of PCR amplification products of *rolB* (381 bp) and *rolC* (408 bp) was observed in randomly selected HRs, while the untransformed normal roots did not show any amplification (Figure 1C,D). Furthermore, the absence of *VirG* genes in the HRs confirmed that the use of antibiotics successfully eradicated *A. rhizogenes* from the culture (Figure 1E). At 14 days, the HRs further elongated and developed dense branches (Figure 1F). Subsequently, 2–3 cm of HRs identified by PCR were excised and cultured in 100 mL of 1/2 MS liquid medium, and the HRs fully occupied the culture vessel by 28 d (Figure 1G).

The effects of density of bacterial suspension, infection time, co-culture medium, and AS on the rate of induction of the HRs were studied to increase the rate of induction of the HRs. As shown in Figure 2A, the induction rate is higher when a density of 0.6, 0.8, or 1.0 OD_600_ is used. This led to an average induction rate that exceeded 40%. When the infection time was 10 and 15 min (Figure 2B), the rates of induction of the HRs were 50.62 ± 2.70% and 43.41 ± 8.95%, respectively, while the rate of induction significantly decreased when the infection time was 5 or 20 min. Figure 2C indicates that the rate of induction of the HRs reaches a higher level of 52.58 ± 7.07% when the co-culture medium is 1/2 MS, while there is no significant difference in the effect of MS, 6,7-V, and B5 on the induction rate of *G. glabra* HRs. The addition of AS, a phenolic compound, to the co-culture medium facilitated the induction of HRs in the plant. Our findings suggested that the addition of 100 μmol/L AS effectively enhanced the efficiency of inducing HRs (Figure 2D). Various factors indicated that the most effective conditions to induce the subsequent HR included bacteria at a density of 0.6 and an infection time of 10 min, followed by transfer to ½ MS medium that contained 100 μmol/L AS for co-cultivation.

### 3.2. Growth Kinetics of G. glabra HRs

The growth kinetics of HRs were studied in 1/2 MS medium over a period of 42 d to identify the proper harvest time. There were few short HR branches during the initial incubation period (7 d). However, the number of HR branches increased significantly at day 14. At 28–35 d, the growth of HR branches reached its peak. By day 42, the growth of HR branches instead stagnated and declined (Figure 3A). We hypothesized that the stagnation of the growth of HRs at 42 d may be owing to the depletion of nutrients in the culture medium and the limited growth space. We measured the biomass of HR throughout the cultivation process, and the results were largely consistent with those of Figure 3A. As the cultivation time increased, the biomass in the shake flasks gradually increased. The biomass increased significantly on day 14. The biomass of the HRs peaked on day 28 and increased from 0.05 g/flask to 1.83 g/flask (Figure 3B). Consistent with the observed results, the FW of HRs in the liquid suspension culture medium increased by 24-fold on day 21 and approached 34.5-fold on day 28 (Figure 3C). The HRs subsequently entered a stable growth phase. The FW remained stable during this phase and did not increase significantly.

### 3.3. Effects of MeJA and SA on the Content of Flavonoids in G. glabra HRs

The ability of MeJA and SA to elicit the biosynthesis of flavonoids was studied by treating 28-day-old *G. glabra* HRs with 100 μM MeJA and 100 μM SA followed by continuous cultivation for 3 d and 5 d. The contents of isoliquiritigenin, liquiritigenin, liquiritin, licochalcone A, and glabridin were analyzed. The levels of isoliquiritigenin and licochalcone A in the HRs increased by 30.25% and 51.08%, respectively, on day 5 of the MeJA treatment (Figure 4A,D). Compared to the control group, the contents of liquiritigenin, liquiritin, and glabridin in HRs significantly increased at all time points following the treatment with MeJA. The content of liquiritigenin in the HRs increased by 59.09% and 100.03% after the MeJA treatment for 3 d and 5 d, respectively (Figure 4B). The content of liquiritin in the HRs increased by 28.23% and 53.55% after treatment with MeJA for 3 d and 5 d, respectively (Figure 4C). In addition, the content of glabridin in the HRs also increased by 23.70% and 36.83% (Figure 4E). After the SA treatment, the content of liquiritin in the HRs increased significantly on both the 3 and 5 days, while the contents of isoliquiritigenin, liquiritigenin, licochalcone A, and glabridin only showed significant increases on day 5 (Figure 4F–J). These results indicated that MeJA was a suitable stimulant to induce the biosynthesis of flavonoids in the *G. glabra* HRs.

### 3.4. GO Association and KEGG Enrichment Analysis in Two Comparisons

The gene regulatory mechanism of MeJA on the biosynthetic pathway of flavonoids in the *G. glabra* HRs was studied using GO classification analyses on all the selected up-regulated and downregulated DEGs in the two groups (0 vs. 3 d, 0 vs. 5 d) (Appendix A). The GO classification analysis categorized the annotated genes into the following three major functional classes: molecular functions, cellular components, and biological processes; the annotated genes encompassed 30 functional groups (Appendix A). The dominant groups between “0 vs. 3 d” within the molecular function, cellular component, and biological process categories were “sequence-specific DNA binding transcription factor activity” (216 members), plasma membrane (448 members), and protein phosphorylation (170 members) (Appendix A). The dominant groups in the categories of molecular function, cellular component, and biological process in the comparison between “0 vs. 5 d” were “sequence-specific DNA binding transcription factor activity” (95 members), extracellular region (96 members), and defense response (64 members), respectively (Appendix A). On days 3 and 5 of the MeJA treatment, 25 and 13 annotated genes were classified into the “flavonoid biosynthetic process”, respectively.

The biological pathways involved in the secondary metabolites were explored further by performing a KEGG pathway enrichment analysis using the upregulated and down-regulated DEGs (0 vs. 3 d, 0 vs. 5 d) to reveal the significantly enriched pathways. After 3 d and 5 d of MeJA treatment, the following KEGG pathways were mostly enriched in “metabolic pathways (ko01100)” (516 DEGs at 3 d and 251 DEGs at 5 d, respectively) and “biosynthesis of secondary metabolites (ko01110)” (343 DEGs at 3 d and 148 DEGs at 5 d, respectively) (Appendix A). Notably, the phenylpropanoid biosynthesis (ko00940), flavonoid biosynthesis (ko00941), isoflavonoid biosynthesis (ko00943), and flavone and flavanol biosynthesis (ko00944) were also significantly enriched in the HRs treated with MeJA, thus indicating their pivotal role in mediating the biosynthesis of flavonoids in the HRs (Appendix A). Additionally, on days 3 and 5, 130 and 41 DEGs potentially related to flavonoid synthesis were identified, respectively (Appendix A and Appendix A). Among them, 65% of the DEGs were upregulated on d 3, while 56% of the genes were up-regulated on d 5.

### 3.5. Expression Clustering Revealed Patterns of Flavonoids Associated with Genes

The dynamics of gene expression in the HRs under different durations of MeJA treatments were studied by performing a K-means clustering analysis on the transcriptomic data collected at various time points. The genes were divided into six clusters (Appendix A and Appendix A). In Cluster 1, the gene expression increased at d 3 and decreased at d 5. In Cluster 2, the gene expression significantly decreased at 3 d and slightly rebounded at 5 d. In Cluster 3, the gene showed no significant change at 3 d but increased significantly at 5 d. In cluster 4, the gene expression exhibited a sustained increase at 3 d and 5 d. The gene expression in cluster 5 significantly decreased on both days 3 and 5, while Cluster 6 showed a significant increase at 3 d and a slight increase at 5 d (Appendix A). A KEGG analysis was employed to focus on the DEGs that were upregulated at both 3 and 5 d, such as the genes in Cluster 4 and Cluster 6 (Appendix A). The genes in Cluster 4 were primarily enriched in the pathways of “Glycolysis/Gluconeogenesis (ko00010)”, “Ga-lactose metabolism (ko00052)”, and “Plant hormone signal transduction (ko04075)”. The genes in Cluster 6 were primarily enriched in the pathways of “Glycolysis/Gluconeogenesis (ko00010)”, “Phenylpropanoid biosynthesis (ko00940)”, and “Flavonoid biosynthesis (ko00941).” In addition, the genes involved in “Phenylalanine (ko00400)” and “Isoflavonoid biosynthesis (ko00943)” were also significantly enriched. These results indicate that the Cluster 6 genes may be involved in the biosynthesis of flavonoids in the HRs after MeJA treatment.

### 3.6. MeJA Effects the Levels of Transcription of the Genes Related to the Biosynthesis of Flavonoids in the G. glabra HRs

The schematic biosynthetic pathway of flavonoids/isoflavonoids and the involved genes are mapped in Figure 5. Numerous studies have shown that MeJA affects the biosynthesis of flavonoids in plants [13,36]. A KEGG enrichment analysis and gene functional annotation revealed that a total of 33 genes and 10 corresponding enzymes were recognized in phenylpropanoid biosynthesis, flavonoid biosynthesis, and isoflavonoid biosynthesis pathways. The sequences and FPKM values are listed in Appendix A. As shown in Figure 6, there were eight significantly upregulated DEGs (1 *PAL*, 3 *CHS*, 1 UGT, 1 *HI4OMT*, 1 *CYP81E*, and 1 *VR*) and four downregulated DEGs (1 *CHS*, 1 *CHI*, and 2 *CYP81E*) in both the 3 d and 5 d. The biosynthesis of flavonoids occurs through the phenylpropanoid metabolic pathway. As the first enzyme in this pathway, phenylalanine ammonia-lyase (PAL) plays a pivotal role in linking the primary metabolism with the phenylpropanoid metabolic pathway [37]. We found that the expression of *GgPAL3* was significantly upregulated on both days 3 and 5 after the MeJA treatment in HRs. 4: Coumarate-CoA ligase (4CL) is a key branch point enzyme in the phenylpropanoid metabolic pathway, and the gene *Gg4CL1* was significantly upregulated on day 3 after the MeJA treatment. CHS is the first key enzyme and rate-limiting enzyme in the flavonoid biosynthetic pathway. It catalyzes the formation of chalcone (precursor of isoliquiritigenin) from coumaroyl-CoA. Here, we found that five *GgCHSs* genes, including *GgCHS1*, *GgCHS3*, *GgCHS6*, and *GgCHS9*, inducing at all time points of the treatment, while *GgCHS7* was significantly downregulated. As previously reported in the literature, many genes in the flavonoid pathway of *Glycyrrhiza* species encode the same enzymes but exhibit different orientations of expression [38,39,40]. This indicates the complexity of the expression of flavonoid biosynthetic genes, and the direction and magnitude of the transcriptional responses of these genes were not conservative. In the biosynthesis of liquiritin, *GgUGT*, which was strongly induced by MeJA, may play a significant role. In the biosynthesis of isoflavones, *GgHI4OMT*, *GgCYP81E4*, and *GgVR1* were significantly upregulated by the MeJA treatment, while *GgCYP81E1* and *GgCYP81E2* were significantly downregulated.

### 3.7. qRT-PCR Validation of Key Genes in the Biosynthesis of Flavonoids

qRT-PCR was used to detect the expression of key genes involved in the biosynthesis of flavonoids in the *G. glabra* HRs after 0, 3, and 5 d of MeJA induction. As shown in Figure 7, the results of qRT-PCR of the selected genes related to the biosynthesis of flavonoids matched well with the transcriptome results. The results showed that the level of expression of *GgPAL*3 exhibited an initial increase followed by a decrease (Figure 7A), indicating that it may play a role during the precursor stage of biosynthesis. As a ligase that connects phenylpropanoid metabolism and flavonoid metabolism, the expression level of expression of *GgCHS6* increased significantly (Figure 7B). In addition, several other genes, including *GgUGT*, *GgHI4OMT*, *GgCYP81E4*, and *GgVR1*, that are involved in the synthesis of flavonoids/isoflavones also exhibited varying degrees of upregulation (Figure 7C–F). The level of expression of *GgCHS7*, *GgCHI1*, and *GgCYP81E2* was largely consistent with the transcriptome data, and it also showed a downward trend under the MeJA treatment (Figure 7G–I).

### 3.8. Determination of the Contents of Flavonoid in GgCHS6 That Overexpressed HRs

Enhancing the content of plant secondary metabolites through genetic engineering is a significant approach in synthetic biology. To further increase the contents of flavonoids in the *G. glabra* HRs, we transformed and obtained HRs that overexpressed *GgCHS6* (Appendix A). As shown in Figure 8A, there was no significant morphological difference between the 1302 (empty vector) HRs and *GgCHS6*-overexpressing HRs. Additionally, there was no notable difference in the FW and growth rate between them (Appendix A). These results indicate that the overexpression of *GgCHS6* does not affect the growth of HRs. The HPLC-MS analysis results indicated that the contents of five flavonoids all increased in HRs overexpressing the *GgCHS6* gene. Compared to the 1302, the contents of isoliquiritigenin, liquiritigenin, liquiritin, and licochalcone A in HRs that overexpressed *GgCHS6* increased by 75.81%, 133.03%, 84.81%, and 67.76%, respectively, while the content of glabridin only increased by 39.74% (Figure 8B–F).

## 4. Discussion

Plant secondary metabolites have garnered attention due to their extensive applications in the fields of healthcare and food processing [41,42]. Over recent decades, plant HRs have emerged as an ideal biological system for both the large-scale biosynthesis of valuable secondary metabolites and the exploration of bioactive compound pathways [7,43]. This study systematically investigated the key parameters that affected the efficiency of inducing HRs (Figure 2). The concentration of *A. rhizogenes* critically influences the induction of HR. The results of this study demonstrated optimal *G. glabra* HR induction at bacterial suspension densities of 0.6–1.0 OD_600_ (Figure 2A). Within this range, the bacterial density correlates proportionally with the induction efficiency, although excessive densities cause browning and necrosis of the explants [7]. Currently, the concentration of *A. rhizogenes* used to induce the formation of roots is typically adjusted to an OD_600_ of 0.6–1.0 [7,17,44]. Similarly to the effect of bacterial suspension density, within a certain range, a longer infection time of the bacterial suspension results in a higher rate of the induction of HRs [45]. Our optimization identified 10 min as the optimal infection time for strain K599, which caused the maximal induction of HRs (Figure 2B). Appropriate co-culture media maintain the viability and cellular activity of the explants, which facilitate the induction of HRs [7]. We confirmed that the 1/2 MS medium was superior at inducing the HRs (Figure 2C). This is consistent with the findings of research on *Astragalus membranaceus* [46]. As a signaling molecule that activates the *rol* genes of *A. rhizogenes*, the phenolic compound AS can also enhance the induction rate of HRs at plant wound sites [7,47]. As expected, the supplementation with AS significantly increased the induction rates, and 100 μM was determined as the optimal concentration (Figure 2D).

The scaled-up cultivation of plant HRs in bioreactors is an essential pathway for the scaling and commercialization of the biosynthesis of secondary metabolites [4,48]. The rapidly growing HRs are crucial for the accumulation of secondary metabolites. Thus, it is imperative to observe the growth kinetic curve of HRs. In this study, selected HR lines showed a 34.5-fold increase in biomass and reached 1.83 g/flask compared to its initial FW (0.05 g/flask) after 28 d (Figure 3). The time course showed that the growth of *G. glabra* HRs followed a 42-day cycle in 100 mL liquid MS medium, including four distinct phases. These included a lag phase (0–14 d), a log phase (14–28 d), a stationary phase (28–35 d), and a phase of decline (after 35–42 d). This phased growth pattern aligns with previous research on *Rubia yunnanensis* Diels [49], *Cannabis sativa* L. [50], and *Polygala tenuifolia* Willd. [51]. These results indicate that under the current shake flask conditions, 28 d is the optimal period to harvest *G. glabra* HRs. Consistent with the observations in *C. sativa* L. [50], *A. membranaceus* [46], and *Dalea purpurea* [52], the growth of HRs plateaus at 28–35 d, and the biomass declines after day 35. In fact, an extension of the growth period depletes the nutrients and limits the growth space within the shake flask or bioreactors; these are the key factors that lead to the stagnation and retardation of plant HRs [53,54].

In plant HR cultures, elicitors are a successful and widely used method to induce the biosynthesis of secondary metabolites. Previous studies have shown that the cellulase from *Aspergillus niger*, MeJA, and *Rhizobium leguminosarum* can significantly increase the content of glycyrrhizin in HRs of *G. glabra* [55,56]. This study demonstrated that treatment with MeJA significantly increased the contents of isoliquiritigenin, liquiritin, and glabridin at all the time points (Figure 4). Thus, the elicitation of MeJA is a promising method to enhance the biosynthesis of flavonoid in *G. glabra* HRs. The transcription analysis enabled observations of the biosynthesis of flavonoids in *G. glabra* HRs under the regulation of MeJA. The GO classification analyses showed that the upregulated and downregulated DEGs were enriched in the following three functional groups: biological process, cellular component, and molecular function (Appendix A). These results indicated that the treatment with MeJA profoundly reprogrammed the biochemistry in *G. glabra* HRs. The KEGG enrichment of upregulated and downregulated DEGs (0 d vs. 3 d and 0 d vs. 5 d) identified metabolic pathways (ko01100) and secondary metabolite biosynthesis (ko01110) as significantly altered pathways (Appendix A). Importantly, the trends of expression of genes involved in phenylpropanoid biosynthesis (ko00940), flavonoid biosynthesis (ko00941), and isoflavonoid biosynthesis (ko00943) were upregulated by MeJA induction (Appendix A), which confirms its pivotal role in mediating the biosynthesis of flavonoids. As a signaling molecule, MeJA has been reported to promote the biosynthesis and accumulation of flavonoids in *Ficus pandurate* [57], *G. inflata* [58], *Dioscorea composita* [59], and *Hedera helix* [60].

Numerous rate-limiting biosynthetic enzymes encoded by multigene families, including *PALs*, *4CLs*, *CHSs*, *CHIs*, and *GTs*, play a key role in the synthesis of flavonoids (Figure 5). This study identified 33 genes and 10 corresponding enzymes within the phenylpropanoid, flavonoid, and isoflavonoid biosynthetic pathways (Figure 6). More importantly, eight genes were significantly upregulated at all the time points following treatment with MeJA. The data of qRT-PCR further confirmed the trend of upregulation of these genes (Figure 7). Consistent with previous studies, treatment with MeJA can promote the upregulation of genes involved in flavonoid biosynthesis in licorice [39,61]. Phenylpropanoid biosynthesis is a major pathway for the biosynthesis of various specialized metabolites to defend against environmental stresses and changes [37]. Treatment with MeJA significantly induced the upregulation of *GgPAL3*, thus suggesting an enhancement of phenylpropanoid metabolism in the HRs, which can provide the precursor chemicals for the subsequent biosynthesis of flavonoids. CHS catalyzes the formation of chalcone from *p*-coumaroyl-CoA and is a key enzyme that directs the phenylpropanoid biosynthesis towards the biosynthesis of flavonoid [62]. MeJA induces the significant upregulation of multiple *CHS* genes, which indicates that MeJA can substantially promote the biosynthesis of flavonoids. More importantly, the overexpression of the *GgCHS6* gene in the HRs of *G. glabra* can also increase the content of flavonoids (Figure 8). Furthermore, the genes involved in the isoflavone biosynthetic pathway, such as *GgHI4OMT*, *GgCYP81E4*, and *GgVR1*, were also significantly upregulated. This indicates that treatment with MeJA promotes the biosynthesis of more downstream metabolites in the HRs. In fact, subsequent research should employ multiple approaches, such as gene editing, gene silencing, or gene overexpression, to further validate the roles of these genes in HRs. Overall, MeJA may induce a regulatory chain that involves *PAL*/*CHS* genes in the biosynthesis of flavonoids in HRs, but its complete and precise regulatory network still merits further study.

## 5. Conclusions

In summary, we established an optimized HR culture system of *G. glabra* for the biosynthesis of various medicinally active ingredients. We identified MeJA as an effective elicitor that can significantly stimulate the accumulation of flavonoids, such as isoliquiritigenin, liquiritigenin, and liquiritin in HRs. The results of a transcriptome analysis revealed that the treatment with MeJA substantially upregulated the flavonoid pathway genes, such as *PAL* and *CHS* in the HRs, which effectively activated the biosynthesis of flavonoids. These findings were further validated by the qRT-PCR data. Additionally, the overexpression of *GgCHS6* in the HRs by genetic engineering significantly increased the content of flavonoids. Collectively, these results elucidate the molecular mechanisms that underlie the flavonoid biosynthesis elicited by MeJA and establish an experimental framework to enhance the production of flavonoids by *G. glabra* through metabolic and genetic engineering. Further research on the genes identified is merited to unravel the complex network of flavonoid biosynthesis in licorice.

## Figures and Tables

**Figure 1 genes-16-01387-f001:**
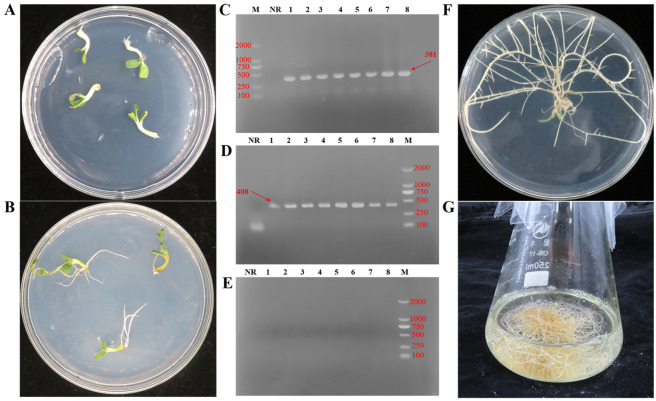
Induction and establishment of *A. rhizogenes* strain K599 mediated HR culture from *G. glabra*. (**A**) At 2 days of co-cultivation, HRs formed at the wound site. (**B**) On the 5th day, the continuously elongating HRs. (**C**) The size of the *rolB* gene is 381 bp. (**D**) The size of the *rolC* gene is 408 bp. (**E**) The size of the *VirG* gene is 319 bp. M: marker; NR: The normal root of licorice. 1–8: Different hairy root samples. (**F**) Highly branched HRs formed in two weeks. (**G**) HRs were grown in a flask containing 1/2 MS liquid medium for 28 d.

**Figure 2 genes-16-01387-f002:**
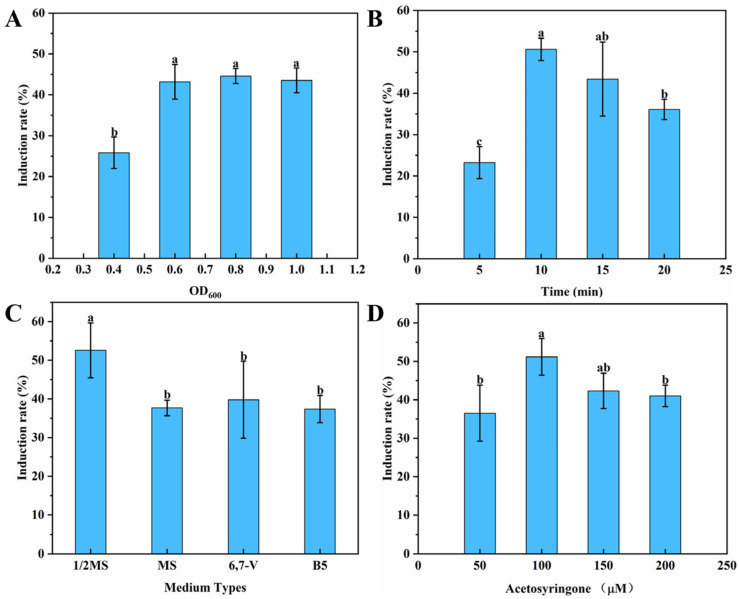
Establishment and optimization of the induction system for *G*. *glabra* HRs. Effects of the density of bacterial suspension on the rate of HR induction in *G. glabra* (**A**), effects of infection time (**B**), types of co-culture medium (**C**), and AS concentration (**D**) on HR induction. The data were shown as the mean ± standard deviation, and the different letters represent significant differences (*p* < 0.05).

**Figure 3 genes-16-01387-f003:**
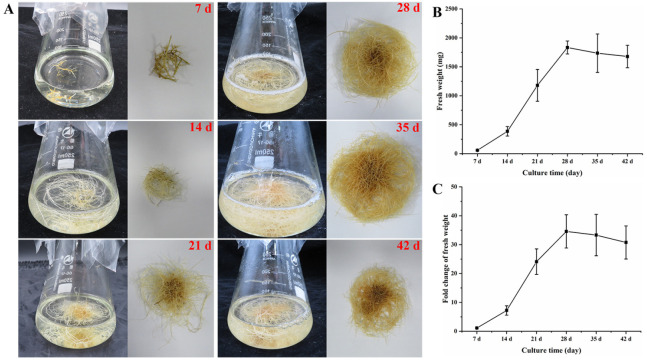
Growth kinetics of *G. glabra* HRs in phytohormone-free 1/2 strength liquid MS medium. (**A**) Change in the growth status of HRs cultured in medium for 42 days. (**B**) The growth curve of fresh weight of HRs at different time periods. (**C**) The curve of fresh weight shows multiple changes in HRs at different time periods. Data were obtained from three independent biological replicates and presented as the mean ± standard deviation.

**Figure 4 genes-16-01387-f004:**
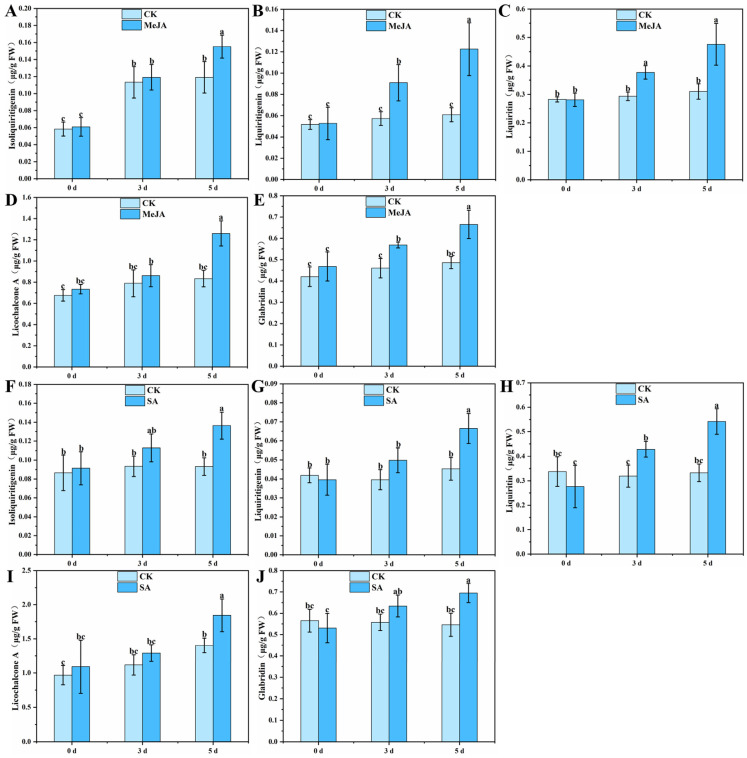
Effects of MeJA and SA on the content of flavonoids in *G. glabra* HRs. (**A**–**E**) Effects of MeJA on the content of licochalcone A, liquiritin, liquiritigenin, isoliquiritigenin, and glabridin in *G. glabra* HRs. (**F**–**J**) Effects of SA on the content of licochalcone A, liquiritin, liquiritigenin, isoliquiritigenin, and glabridin in *G. glabra* HRs. The data were shown as the mean ± standard deviation, and the different letters represent significant differences (*p* < 0.05).

**Figure 5 genes-16-01387-f005:**
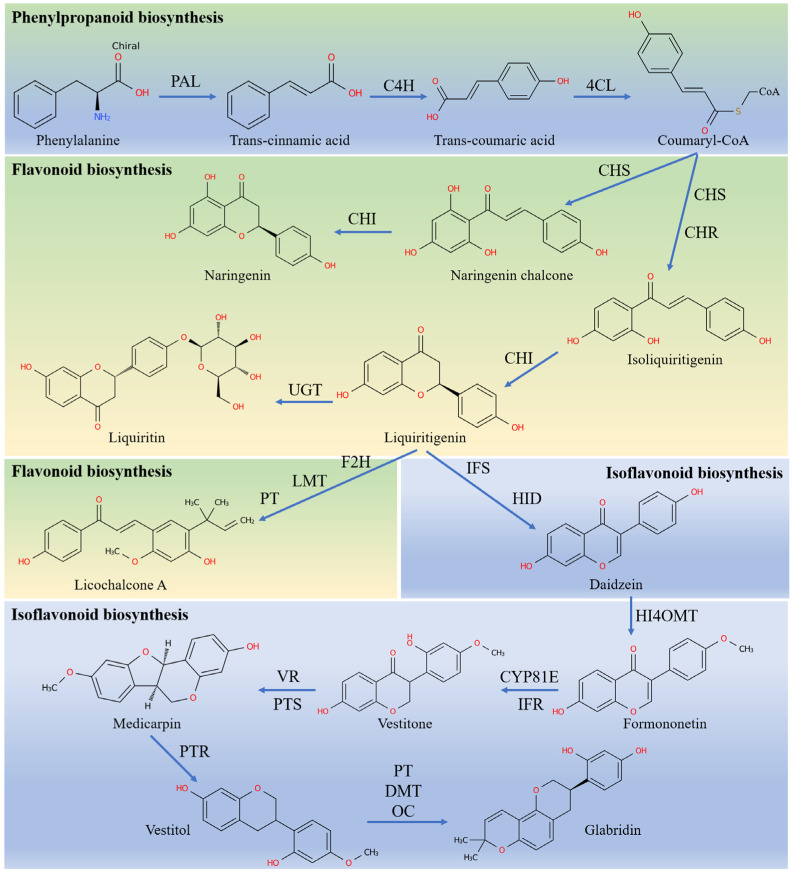
Expression profiles of enzyme-encoding genes involved in phenylpropanoid and flavonoid/isoflavone biosynthesis. Abbreviations: PAL (phenylalanine ammonialyase), C4H (cinnamate 4-hydroxylase), 4CL (4-coumaroyl CoA ligase), CHS (chalcone synthase), CHR (chalcone reductase), CHI (chalcone isomerase), UGT (UDP-glucosyltransferase), F2H (flavanone 2′-hydroxylase), LMT (licodione 2′-O-methyltransferase), IFS (2-hydroxyisoflavanone synthase), HID (2-hydroxyisoflavanone dehydratase), HI4OMT (isoflavone 4′-O-methyltransferase), CYP81E (isoflavone 2′-hydroxylase), IFR (isoflavone reductase), VR (vestitone reductase), PTS (pterocarpan synthase), PTR (pterocarpan reductase), PT (prenyltransferases), DMT (demethylases), and OC (Isopentenyl cyclase).

**Figure 6 genes-16-01387-f006:**
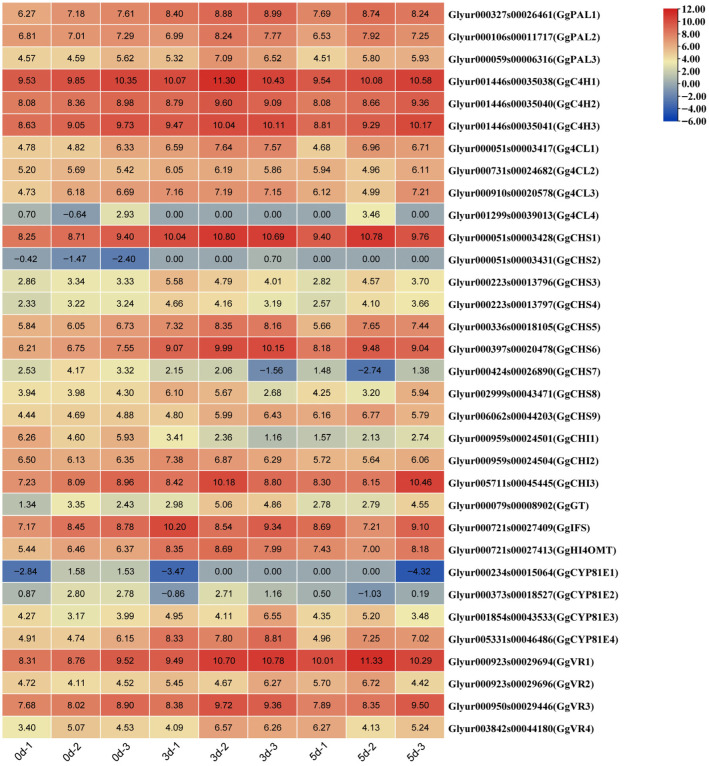
The heat graph shows the up-regulated and down-regulated genes involved in phenylpropanoid and flavonoid/isoflavone biosynthesis in *G. glabra* HRs under MeJA treatment. The heatmap values represent the Log_2_FPKM value of each sample.

**Figure 7 genes-16-01387-f007:**
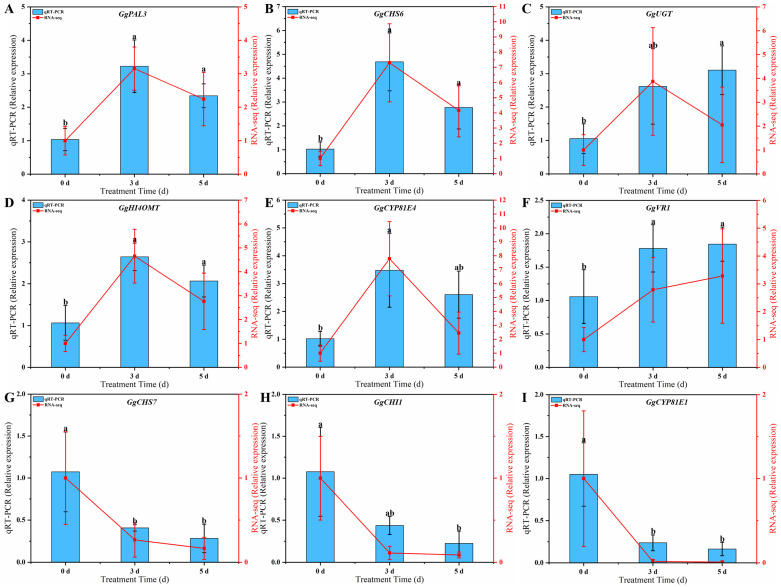
The qRT-PCR analysis of selected DEGs in *G. glabra* HRs under the MeJA treatment. *GgPAL3* (**A**), *GgCHS6* (**B**), *GgUGT* (**C**), *GgHI4OMT* (**D**), *GgCYP81E4* (**E**), *GgVR1* (**F**), *GgCHS7* (**G**), *GgCHI1* (**H**), *GgCYP81E1* (**I**). Column diagram represents the relative expression level of qRT-PCR (left y-axis). The line chart represents the relative expression level of RNA-Seq (right y-axis). Different small letters in the figure showed a significant difference (*p* < 0.05).

**Figure 8 genes-16-01387-f008:**
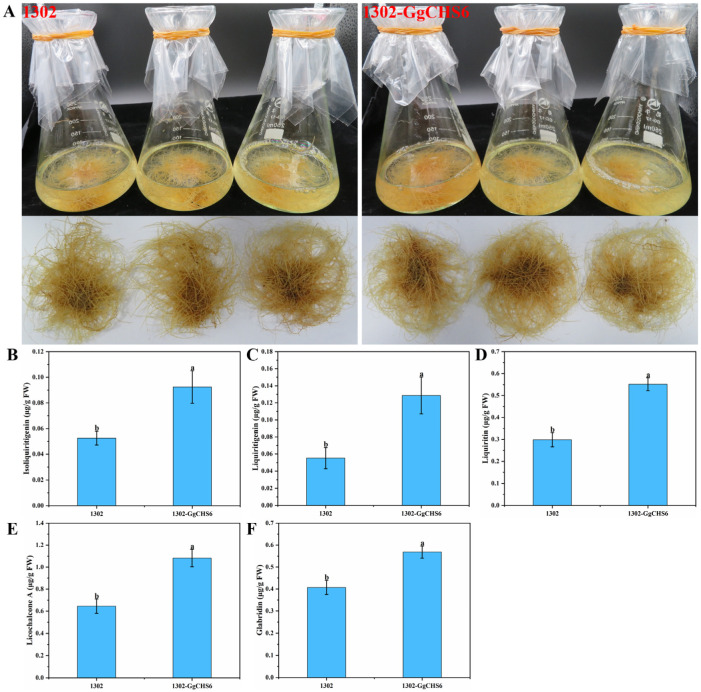
The impact of *GgCHS6* overexpression on *G. glabra* HRs. (**A**) The impact of *GgCHS6* gene overexpression on the morphology of HRs. (**B**–**F**) The overexpression of the *GgCHS6* gene affects the content of isoliquiritigenin (**B**), liquiritigenin (**C**), liquiritin (**D**), licochalcone A (**E**), and glabridin (**F**). The data were shown as the mean ± standard deviation, and the different letters represent significant differences (*p* < 0.05).

## Data Availability

Data is contained within the article and Appendix A.

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
