# Peer review of "MeJA Elicitation on Flavonoid Biosynthesis and Gene Expression in the Hairy Roots of Glycyrrhiza glabra L."

_genes, 2025, doi:10.3390/genes16111387_

Round 1
Reviewer 1 Report
Comments and Suggestions for Authors
The authors of the manuscript ‘Insight into the genetic mechanism in flavonoid biosynthesis triggered by MeJA in Glycyrrhiza glabra L. hairy roots’ developed a protocol for hairy roots culture of Glycyrrhiza glabra and conducted transcriptome analysis to identify gene expression changes after treatment with methyl jasmonate. The manuscript is well prepared; however, the authors overlooked previous studies on this species. Numerous reports are available on metabolite production through the elicitation (biotic and abiotic elicitors) of hairy roots of Glycyrrhiza glabra.
Afsharzadeh, N., Paltram, R., Jungwirth, A., Tabrizi, L., Nazeri, V., Kalantari, H., Halbwirth, H., Samiei, L., Sheehan, H. and Shokrpour, M., 2025. Advancing Glycyrrhiza glabra L. Cultivation and Hairy Root Transformation and Elicitation for Future Metabolite Overexpression. Horticulturae, 11(1), p.62.
Allahdou, M., Mehravaran, L. and Khajeh, H., 2025. Improving the Production of Glycyrrhizin and Glycyrrhetinic Acid by Eliciting Hairy Root of Glycyrrhiza glabra. Russian Journal of Plant Physiology, 72(1), p.3.
Behdad, A. and Ganjeali, A., 2025. Phytohormones and microbial elicitation on glycyrrhizin production and gene expression in the hairy root of Glycyrrhiza glabra L. Plant Cell, Tissue and Organ Culture (PCTOC), 160(1), p.4.
The authors should incorporate recent studies into the introduction. Discuss results such as hairy root induction, biomass, and metabolite production in relation to previous research to emphasize the significance of this study.
Author Response
On behalf of my co-authors, we thank you very much for allowing us to revise our manuscript. We have studied the comments carefully and have made revisions which are marked in red in the new manuscript. We have tried our best to revise our manuscript according to the comments. Attached please find the revised version, which we would like to submit for your kind consideration.
We sincerely appreciate the reviewers' comments and have supplemented the omitted references in both the Introduction and Discussion sections as requested.

Reviewer 2 Report
Comments and Suggestions for Authors
The authors report that Glycyrrhiza glabra produces more flavonoids if treated with MeJA. This is known for many other plant species and thus not a novel findings. Unfortunately, this study does not reveal any mechanistic insights. Please find my specific comments below.
- line 17: "etc.," does not make sense, because readers will not know this.
- "which possess pharmacological activities such as anti-tumor, anti-viral, antioxidant, and immunomodulatory effects" ... these are very strong medical claims that lack evidence. I would recommend to just remove this.
- The abstract already reveals that the authors only conducted superficial analyses that revealed correlations, but not mechanistic insights as suggested by the title. Therefore, the title needs to be corrected to accurately reflect the actual content of the manuscript.
- "clearing heat and detoxifying, alleviating spasms and pain, resolving phlegm and relieving cough, and harmonizing the effects of other medicines [2,3]." ... more medical claims without the necessary evidence. The cited references are just (low quality) review articles. Properly conducted clinical trials would be needed to infer these claims.
- "competition for arable land between medicinal and food crops" ... I am not aware of this competition and do not believe that it exists. If medicinal plants are eaten, they also help against hunger.
- "In the present study, the addition of MeJA can promote the increase in the contents of isoliquiritigenin, liquiritigenin, liquiritin, licochalcone" ... would be sad if this would not be generally the case.
- It is strange that the authors work with transgenic plants, but do not use this opportunity to overexpress the flavonoid biosynthesis by introducing the necessary transcription factors (see for example MYB+bHLH overexpression, 10.1038/nbt.1506 ).
- If there is already a genome sequence available, the de novo transcriptome assembly does not make sense. Gene expression could be studied based on the available genome sequence.
- The critical point is missing in the methods: identification of the genes involved in the flavonoid biosynthesis. There are multiple gene copies for many steps thus the correct identification of the functional copy e.g. via KIPEs would be crucial. Also, identification of the correct transcription factors would be essential.
- Data analysis: What is the reason for displaying standard error instead of standard deviation? Also, the statement about SPSS contradicts the previous paragraph about DESeq2.
- Fig.2 suggest that there are significant differences between different tested conditions. However, the effect size looks extremely small. Is there really a practical relevance of these findings?
- Fig.3: Again, I believe that the standard deviation should be displayed here0.
- The use of FPKM + FC to find differentially expressed genes contradicts the method section that stated DESeq2.
- Section 3.4 does not reveal any biological results. Numerous numbers are presented, but what does this mean?
- GO and KEGG enrichment analyses are usually too superficial to be useful. Here, the authors should try to understand how the specific flavonoids are produced that are present in high abundance. Therefore, it is even more important to look at individual genes and not at the pathway level. It is already clear that the flavonoid biosynthesis must be enriched.
- Fig.5-8 are all connected to these enrichment analyses and should be moved to the supplements. These figures do not reveal biological insights.
- line 382-line391 could be omitted.
- The KEGG annotation is superficial resulting in multiple genes like 3x or 5x (?) CHS. Are all these copies really functional CHS? This is where a proper annotation with KIPEs (10.1371/journal.pone.0294342) would help.
- Fig10 could also show the expression values as numbers written into the individual fields. Also, the different replicates can be shown as individual columns. There is plenty of space for a wider figure.
- RT-qPCR is not a validation of RNA-seq. It is just a different method. If RNA-seq was conducted properly, there is no need for qPCR. Also, it seems that the scale of the two y-axis differs between individual gene plots in Fig.11. This makes all plots look like the results would be consistent. However, only very general patterns might match between the genes.
- line 471-480 reads like a second introduction and should be removed to make the manuscript more concise.
- "Scale-up cultivation of plant HRs in shake flasks is an essential pathway for the scaling and commercialization of secondary metabolite biosynthesis" ... is this really the best approach? I would recommend to go for bioreactors instead.
- "Then, the molecular mechanisms of flavonoid biosynthesis in G. glabra HRs under the regulation of MeJA were revealed through transcriptome analysis." ... this is not accurate. Some observations were made, but no mechanisms were revealed.
- Identification of the bottlenecks in the biosynthesis would make sense, but that would require more in-depth analyses. CHS is a committed step in the flavonoid biosynthesis thus overexpression should have a positive impact. However, fine tuning competing branches within the flavonoid biosynthesis would be very promising too.
- "rapid synthesis of various medicinal active ingredients" ... no, this claim is not supported by the necessary evidence.
- "We identified MeJA as an optimal elicitor" ... no, this was the only tested elicitor in this study. A comparison of all possible elicitors would be required to arrive at this conclusion.
- This study involved RNA-seq datasets which are obviously not contained in the Supplementary Material. The reads need to be submitted to the Sequence Read Archive.
Author Response
On behalf of my co-authors, we thank you very much for allowing us to revise our manuscript. We have studied the comments carefully and have made revisions to the paper. We have tried our best to revise our manuscript according to the comments. Attached please find the revised version, which we would like to submit for your kind consideration.
To point 1: line 17: "etc.," does not make sense, because readers will not know this.
We sincerely appreciate the reviewers' professional comments. We have revised this section in the new manuscript and conducted a thorough check for similar errors throughout the entire text.
To point 2: "which possess pharmacological activities such as anti-tumor, anti-viral, antioxidant, and immunomodulatory effects" ... these are very strong medical claims that lack evidence. I would recommend to just remove this.
We sincerely appreciate the reviewers' professional comments. We have removed this section in the new manuscript.
To point 3: The abstract already reveals that the authors only conducted superficial analyses that revealed correlations, but not mechanistic insights as suggested by the title. Therefore, the title needs to be corrected to accurately reflect the actual content of the manuscript.
We sincerely appreciate the reviewers' constructive comments, which are crucial for improving the manuscript quality. Based on the reviewers' suggestions and the content of the manuscript, we have revised the new title to "MeJA elicitation on flavonoid biosynthesis and gene expression in the hairy roots of Glycyrrhiza glabra L.".
To point 4: "clearing heat and detoxifying, alleviating spasms and pain, resolving phlegm and relieving cough, and harmonizing the effects of other medicines [2,3]." ... more medical claims without the necessary evidence. The cited references are just (low quality) review articles. Properly conducted clinical trials would be needed to infer these claims.
We sincerely appreciate the reviewers' comments. To ensure the reliability of the article's descriptions, these citations lacking direct evidence have been removed from the new manuscript.
To point 5: "competition for arable land between medicinal and food crops" ... I am not aware of this competition and do not believe that it exists. If medicinal plants are eaten, they also help against hunger.
We sincerely appreciate the reviewers' comments. We have removed these ambiguous descriptions in the revised manuscript.
To point 6: "In the present study, the addition of MeJA can promote the increase in the contents of isoliquiritigenin, liquiritigenin, liquiritin, licochalcone" ... would be sad if this would not be generally the case.
We sincerely appreciate the reviewers' comments. In this study, we employed LC-MS to determine the content of relevant compounds. The results demonstrated that MeJA exhibited certain promotive effects on the levels of isoliquiritigenin, liquiritigenin, liquiritin, and glabridin in the hairy roots.
To point 7: It is strange that the authors work with transgenic plants, but do not use this opportunity to overexpress the flavonoid biosynthesis by introducing the necessary transcription factors (see for example MYB+bHLH overexpression, 10.1038/nbt.1506).
We sincerely appreciate the reviewers' comments. The failure to identify and incorporate transcription factors is indeed a significant limitation of this study. In subsequent research, we will focus on the transcription factors regulating flavonoid biosynthesis to further elucidate the mechanisms underlying flavonoid biosynthesis.
To point 8: If there is already a genome sequence available, the de novo transcriptome assembly does not make sense. Gene expression could be studied based on the available genome sequence.
We sincerely appreciate the reviewers' suggestions and comments. At the time of conducting this work, the genome of Glycyrrhiza glabra had not been sequenced. Therefore, we adopted a de novo assembly strategy using Glycyrrhiza uralensis genome as the reference.
To point 9: The critical point is missing in the methods: identification of the genes involved in the flavonoid biosynthesis. There are multiple gene copies for many steps thus the correct identification of the functional copy e.g. via KIPEs would be crucial. Also, identification of the correct transcription factors would be essential.
We sincerely appreciate the reviewers' suggestions and comments. Neglecting the issue of copy genes was indeed a significant oversight, and future in-depth work will give full consideration to this matter. Meanwhile, for the crucial transcription factors, we will also prioritize further exploration in subsequent studies.
To point 10: Data analysis: What is the reason for displaying standard error instead of standard deviation? Also, the statement about SPSS contradicts the previous paragraph about DESeq2.
We sincerely appreciate the reviewers' suggestions and comments. We apologize for mistakenly presenting standard deviation as standard error, and we have corrected this in the revised manuscript.
We sincerely appreciate the reviewers' suggestions and comments. In the article, SPSS was used for analyzing the statistical significance of data, with p<0.05 set as the threshold for determining significant differences. DESeq2, on the other hand, was employed for differential gene expression analysis in transcriptomics. These two methods analyze different types of data.
To point 11: Fig.2 suggest that there are significant differences between different tested conditions. However, the effect size looks extremely small. Is there really a practical relevance of these findings?
We sincerely appreciate the reviewers' suggestions and comments. Although the induction rates of hairy roots showed little difference under various test conditions, optimizing the induction conditions is necessary to save experimental costs and time.
To point 12: Fig.3: Again, I believe that the standard deviation should be displayed here.
We sincerely appreciate the reviewers' suggestions and comments. We apologize for mistakenly presenting standard deviation as standard error, and we have corrected this in the revised manuscript.
To point 13: The use of FPKM + FC to find differentially expressed genes contradicts the method section that stated DESeq2.
We sincerely appreciate the reviewer for pointing out this error. We have made the corresponding revisions in the new manuscript.
To point 14: Section 3.4 does not reveal any biological results. Numerous numbers are presented, but what does this mean?
We sincerely appreciate the reviewer for raising this question. We sincerely appreciate the reviewers' suggestions and comments. Considering the limited practical significance of this paragraph, we have removed this section in the revised manuscript.
To point 15: GO and KEGG enrichment analyses are usually too superficial to be useful. Here, the authors should try to understand how the specific flavonoids are produced that are present in high abundance. Therefore, it is even more important to look at individual genes and not at the pathway level. It is already clear that the flavonoid biosynthesis must be enriched.
We sincerely appreciate the reviewers' comments and suggestions. In this study, we further validated whether MeJA activates the molecular network of flavonoid biosynthesis through GO annotation and KEGG analysis. Subsequently, potential key genes involved in flavonoid synthesis were identified via K-means clustering analysis and gene expression heatmaps, and the functions of these genes were further verified through gene overexpression.
To point 16: Fig.5-8 are all connected to these enrichment analyses and should be moved to the supplements. These figures do not reveal biological insights.
We sincerely appreciate the reviewers' comments and suggestions. As requested, we have moved the corresponding figures to the appendix.
To point 17: line 382-line391 could be omitted.
We sincerely appreciate the reviewers' suggestions. We have removed this section in the new manuscript.
To point 18: The KEGG annotation is superficial resulting in multiple genes like 3x or 5x (?) CHS. Are all these copies really functional CHS? This is where a proper annotation with KIPEs (10.1371/journal.pone.0294342) would help.
We sincerely appreciate the reviewers' comments and suggestions. In our subsequent research, we will employ this tool for detailed analysis to uncover more useful genes.
To point 19: Fig10 could also show the expression values as numbers written into the individual fields. Also, the different replicates can be shown as individual columns. There is plenty of space for a wider figure.
We sincerely appreciate the reviewers' suggestions. We have revised the figure according to the reviewers' comments.
To point 20: RT-qPCR is not a validation of RNA-seq. It is just a different method. If RNA-seq was conducted properly, there is no need for qPCR. Also, it seems that the scale of the two y-axis differs between individual gene plots in Fig.11. This makes all plots look like the results would be consistent. However, only very general patterns might match between the genes.
We sincerely appreciate the reviewers' suggestions. To avoid potential misunderstanding among readers, we have removed the statement "RT-qPCR is not a validation method for RNA-seq" from the revised manuscript. Regarding the scale differences in the Y-axis of individual gene expression graphs in Figure 11, it should be noted that quantitative fluorescence and RNA-seq measurements rarely yield identical numerical values for relative gene expression levels. The consistent upregulation/downregulation trends observed between the transcriptome data across different time points and the qPCR results sufficiently determine the actual gene expression patterns.
To point 21: line 471-480 reads like a second introduction and should be removed to make the manuscript more concise.
Thank you very much for your suggestions. We have deleted that section in the new manuscript to ensure the conciseness of the article.
To point 22: "Scale-up cultivation of plant HRs in shake flasks is an essential pathway for the scaling and commercialization of secondary metabolite biosynthesis" ... is this really the best approach? I would recommend to go for bioreactors instead.
I apologize for our incorrect description. We have revised that section in the new manuscript to ensure the accuracy of the article.
To point 23: "Then, the molecular mechanisms of flavonoid biosynthesis in G. glabra HRs under the regulation of MeJA were revealed through transcriptome analysis." ... this is not accurate. Some observations were made, but no mechanisms were revealed.
We sincerely appreciate the reviewers' suggestions. In the revised manuscript, we have modified this section to ensure the accuracy of the article's description.
To point 24: Identification of the bottlenecks in the biosynthesis would make sense, but that would require more in-depth analyses. CHS is a committed step in the flavonoid biosynthesis thus overexpression should have a positive impact. However, fine tuning competing branches within the flavonoid biosynthesis would be very promising too.
We sincerely appreciate the reviewers' comments. The biosynthesis of flavonoids is an extremely complex process, and further research is needed to refine their synthetic network.
To point 25: "rapid synthesis of various medicinal active ingredients" ... no, this claim is not supported by the necessary evidence.
We sincerely appreciate the reviewers' comments. Indeed, there is a lack of sufficient evidence to support the claim of rapid synthesis of various medicinal active ingredients using hairy roots. In the revised manuscript, we have made corrections by removing the term "rapid" to ensure the accuracy of the article's description.
To point 26: "We identified MeJA as an optimal elicitor" ... no, this was the only tested elicitor in this study. A comparison of all possible elicitors would be required to arrive at this conclusion.
We sincerely appreciate the reviewers' comments. To ensure the rigor of the article, we have changed the term "optimal" to "effective".
To point 27: This study involved RNA-seq datasets which are obviously not contained in the Supplementary Material. The reads need to be submitted to the Sequence Read Archive.
We sincerely appreciate the reviewers' comments. The raw sequence data reported in this paper have been deposited in the Genome Sequence Archive in National Genomics Data Center (Nu-cleic Acids Res 2022), China National Center for Bioinformation/Beijing Institute of Genomics, Chinese Academy of Sciences (GSA: CRA029106) that are publicly accessible at https://ngdc.cncb.ac.cn/gsa. Since the transcriptome data has not yet been officially released in the database, it is currently unavailable for retrieval. Once the database releases the transcriptome data, readers will be able to download it normally.
To point 28: Fig10 could also show the expression values as numbers written into the individual fields. Also, the different replicates can be shown as individual columns. There is plenty of space for a wider figure.
We sincerely appreciate the reviewers' suggestions. We have revised the figure according to the reviewers' comments.
